# Reduced Granule Cell Proliferation and Molecular Dysregulation in the Cerebellum of Lysosomal Acid Phosphatase 2 (ACP2) Mutant Mice

**DOI:** 10.3390/ijms22062994

**Published:** 2021-03-15

**Authors:** Xiaodan Jiao, Maryam Rahimi Balaei, Ejlal Abu-El-Rub, Filippo Casoni, Hassan Pezeshgi Modarres, Sanjiv Dhingra, Jiming Kong, Giacomo G. Consalez, Hassan Marzban

**Affiliations:** 1Department of Human Anatomy and Cell Science, The Children’s Hospital Research Institute of Manitoba (CHRIM), Rady Faculty of Health Sciences, Max Rady College of Medicine, University of Manitoba, Winnipeg, MB R3E 0W2, Canada; JIAO.XIAODAN@umanitoba.ca (X.J.); rahimibm@myumanitoba.ca (M.R.B.); Jiming.Kong@umanitoba.ca (J.K.); 2Department of Physiology and Pathophysiology, Institute of Cardiovascular Sciences, St. Boniface Hospital Research Centre, Rady Faculty of Health Sciences, Max Rady College of Medicine, University of Manitoba, Winnipeg, MB R3E 0W2, Canada; eabuelrub@sbrc.ca (E.A.-E.-R.); Sanjiv.Dhingra@umanitoba.ca (S.D.); 3Physiology and Pathophysiology, Department of Basic Medical Sciences, Faculty of Medicine, Yarmouk University, Irbid 21163, Jordan; 4Division of Neuroscience, San Raffaele Scientific Institute, San Raffaele University, 20132 Milan, Italy; casoni.filippo@hsr.it (F.C.); g.consalez@hsr.it (G.G.C.); 5BioMEMS and Bioinspired Microfluidic Laboratory, Department of Mechanical and Manufacturing Engineering, University of Calgary, Calgary, AB T2N 1N4, Canada; hassan.pezeshki@gmail.com

**Keywords:** granule cells, SHH, MYCN, cerebellum, mice, *nax*

## Abstract

Lysosomal acid phosphatase 2 (*Acp2*) mutant mice (naked-ataxia, *nax*) have a severe cerebellar cortex defect with a striking reduction in the number of granule cells. Using a combination of in vivo and in vitro immunohistochemistry, Western blotting, BrdU assays, and RT-qPCR, we show downregulation of MYCN and dysregulation of the SHH signaling pathway in the *nax* cerebellum. MYCN protein expression is significantly reduced at P10, but not at the peak of proliferation at around P6 when the number of granule cells is strikingly reduced in the *nax* cerebellum. Despite the significant role of the SHH–MycN pathway in granule cell proliferation, our study suggests that a broader molecular pathway and additional mechanisms regulating granule cell development during the clonal expansion period are impaired in the *nax* cerebellum. In particular, our results indicate that downregulation of the protein synthesis machinery may contribute to the reduced number of granule cells in the *nax* cerebellum.

## 1. Introduction

The cerebellum fulfills important roles in motor control, language, attention, and mental imagery [1,2,3]. This brain region has a three-layered cortex that consists of the molecular layer (ML), Purkinje cell (PC) layer, and granule cell (GC) layer [4,5]. Out of the several types of neurons found in these layers, GCs are the most prominent, comprising approximately 80% of neurons in the central nervous system (CNS) [6,7].

During cerebellar development, GC precursors arise from the cerebellar rhombic lip and migrate rostrolaterally in a subpial stream to form the external germinal zone (EGZ) from embryonic day (E) 12.5 to 17 [4,8,9,10]. GC precursors rapidly proliferate in the EGZ and, after 7–8 divisions (~128–144 h), move radially to their final destination and differentiate into mature GCs [11,12,13,14]. During this process, sonic hedgehog (SHH), secreted by PCs, promotes proliferation of GC precursors [15,16,17,18]. It has been suggested that SHH signaling pathway effectors may be involved in GC differentiation as well [19,20]. The SHH receptor Patched 1 (PTCH1), together with the co-receptors CDON (Codon), BOC (Brother of Codon), and GAS1, functions as a control switch that turns on SHH signal transduction [20,21,22]. Upon SHH binding to its receptors, PTCH1-mediated inhibition of SMO (Smoothened; a G-coupled transmembrane receptor) is lifted, allowing constitutive SMO activity and associated downstream signaling [23].

The Gli (glioma-associated oncogene homolog) family of proteins plays an important role in downstream SHH signaling [24,25]. The *Gli* transcription factors include three proteins encoded by the genes *Gli1*, *-2*, and *-3* [26]. The zinc finger domains in GLI proteins bind to DNA to initiate or inhibit transcription. Activation of the SHH pathway facilitates the accumulation of GLI1 in the nucleus, leading to the activation of target genes [27]. GLI2 and -3 have been shown to act as activators or inhibitors of transcription, depending on whether or not they have been converted into their repressor (Gli2-R, Gli3-R) forms [28].

Among other GLI target genes, *Mycn*, located on chromosome 12 in the mouse and encoding the MYCN protein, promotes proliferation of cerebellar GC precursors [5,29,30]. In addition to regulating the proliferative cycle of GC precursors, a decrease in MYCN has been reported to act as a switch from proliferation to differentiation [5,30]. Furthermore, GLI factors target cyclins D and E, which together are thought to promote the proliferation of GC precursors in the EGZ [31,32,33,34]. 

Lysosomal acid phosphatase (Acp2), one of many soluble luminal hydrolases, hydrolyzes orthophosphoric monoesters into alcohol and phosphate and removes the mannose-6-phosphate recognition marker from lysosomal proteins [35,36]. An *Acp2* mutant mouse called *nax* (naked and ataxia), resulting from a spontaneous point mutation in the *Acp2* gene, exhibits an overall impairment of cerebellar cortex development and severe defects in cerebellar function [37]. We have shown that ACP2 expression marks the caudal midbrain and cerebellum [38], and the anterior cerebellum is the most severely affected in the *nax* mice [39], which indicate critical and specific roles in cerebellar development. In this mutant, the highly organized three-layered cerebellar cortex is disrupted and the number of cerebellar GCs is significantly reduced [37,39]. In addition, it has recently been shown that dysregulation of the SHH signaling pathway in animal models with lysosomal disorders are implicated in the proliferation of cerebellar GC precursors [40,41]. 

Based on our current study, we report that the SHH–MycN pathway is impaired in the *nax* cerebellum, but only after the peak of GC proliferation, between postnatal days P4 and P8 [42,43]. In the *nax* cerebellum, defective GC proliferation may be associated with downregulation of the protein synthesis machinery. 

## 2. Results

### 2.1. The Number of Cerebellar GCs Is Significantly Decreased in the nax Cerebellum

Previously, we reported that the *nax* mouse is smaller in size than the wt and shows an overall impairment of the cerebellar cortex and nuclei. The orderly arrangement of the three-layered cerebellar cortex is disrupted in the *nax* cerebellum with a significant decrease in GCs and excessive migration of PCs [38,39].

In order to assess the differences in the number of cerebellar GCs between the wt and *nax* cerebella, the neural nuclei (NeuN) antibody was used to label mature GCs. At P13, NeuN immunostaining shows a large number of GCs densely packed in the GC layer of the wt cerebellum (Figure 1A–C), which could not be observed in the *nax* cerebellum (Figure 1D–F). Further analysis revealed that the number of cerebellar GCs in *nax* mice is significantly reduced to roughly 20% of that scored in wt littermates (Figure 1G).

This dramatic drop in GCs could be due to a defect in proliferation and/or defective migration from the EGZ (aka external GC layer (EGL)) to the definitive GC layer [44]. To explore whether the proliferative activity of GC precursors is impaired in the EGZ, immunostaining for paired box protein 6 (PAX6) was used to label GC precursors in wt and *nax* cerebella at P6, corresponding to the peak of GC precursor clonal expansion. Our results revealed that the number of GC precursors is dramatically decreased in the *nax* EGZ (Figure 2A,B) and is only about 20% of that in wt littermates (Figure 2C). In order to further confirm this observation and identify GC precursors in the EGZ in relation to PC development, we applied double staining for PAX6 and the calcium-binding protein calbindin (CALB), a marker for PCs, in cerebellar sections. At P8, the wt cerebellum shows a thick EGZ containing GC precursors in the medial and lateral cerebellum (Figure 2D,F,G; ~10–12 layers of the PAX6^+^ cells). In contrast, the *nax* mutant cerebellum displays a reduced number of GC precursors in the EGZ, which is hardly detectable near the midline (putative vermis) and very thin in the lateral cerebellum (Figure 2E,H,I; ~ 1–4 layers of the PAX6^+^ cells).

Bergmann glial cells are involved in GC migration, the original concept of glial-guided neuronal migration [45]. Glial fibrillary acidic protein (GFAP) immunostaining in wt (Appendix A) and *nax* (Appendix A) cerebella showed that Bergmann glial cell bodies are present in both groups but are scattered in the PC/molecular layer in the *nax* cerebellum. This can likely be explained by the observation that the multilayer PCs stomata in the molecular layer and Bergmann glial fibers at different lengths are still extending toward the pia mater. To examine whether the proliferating GC precursors migrate to the GC layer, BrdU incorporation was studied. BrdU administration at E18 and subsequent immunostaining at P5 revealed that GCs migrated from the EGZ to the granular layer in the *nax* cerebellum (Figure 3B; arrows in inset), comparable to wt littermates (Figure 3A; arrows in inset). NeuN immunostaining of serial sections confirmed the presence of mature GCs in the granular layer of the wt (Figure 3C) and *nax* cerebellum (Figure 3D). To determine whether GC precursor proliferation is decreased in the *nax* cerebellum, the mice were injected with BrdU at P6 and studied 45 min after injection (Figure 4). BrdU immunostaining revealed that GC proliferation is dramatically decreased in the cerebella of *nax* mice to roughly 30% (Figure 4B,D,G) when compared with wt littermates (Figure 4A,C,G). To further confirm, we performed Ki67 immunohistochemistry on cerebellar sections at P6, demonstrating the abundant presence of Ki67 immunopositive cells in the EGZ of wt mice, indicative of a proliferative environment (Figure 4E,H). Supporting our BrdU results, the number of proliferating (Ki67 positive) cells was markedly lower (to ~ 25%) in the *nax* cerebellum (Figure 4F,H). 

### 2.2. Is the SHH Signaling Pathway Impaired in the nax Cerebellum?

The SHH signaling pathway is an important regulator of GC precursor proliferation during cerebellar development [16,23,24]. To determine whether the SHH pathway is impaired in *nax* mice and if this could underpin disrupted GC proliferation, section immunostaining and Western blotting were performed using wt and *nax* cerebella. PCs represent the primary source from which SHH is secreted. Immunostaining of P12 cerebellar sections revealed that *nax* PCs express SHH at levels comparable to those of wt PCs (Figure 5A,B; Appendix A (the secondary antibody control)). To precisely assess the dynamics of SHH protein expression over time, Western blotting was performed using samples harvested at P5, -7, -10, -15 and -19 from wt and *nax* cerebella. SHH protein levels peaked at P10-P15 in the wt mouse and declined thereafter (Figure 5C,D). Interestingly, in the *nax* cerebellum, SHH expression appears to increase beyond P10 and is maintained at high levels through P19 (Figure 5C,D).

SHH signaling is regulated by multiple mechanisms and interactions with various proteins [46]. Extracellularly, the SHH signaling response and distribution are regulated by HHIP, CDON (Codon), BOC (Brother of Codon), and GAS1, which dynamically interact with the Patched 1 (receptor), leading to the controlled release of SMO inhibition [46]. RT-qPCR showed that *Hhip* gene expression in the *nax* cerebellum was comparable to control levels at P5 and P10 (Figure 6A). At P15, however, an increase in *Hhip* gene expression was observed in wt, whereas levels remained low and comparable to P5 and 10 in the *nax* cerebellum (Figure 6A). *Ptch1* gene expression did not differ significantly from controls at any time-point (Figure 6B).

The GLI family of zinc finger transcription factors is a downstream nuclear effector of the SHH signaling pathway. GLI proteins participate in cell fate determination, proliferation and patterning of many cell types during embryonic development [25]. *Gli1–3* expression was assessed by RT-qPCR in *nax* and wt littermate cerebella (Figure 6C–E). As compared to P5, *Gli1* and *-2* gene expression decreased over time at P10 and 15; no differences in *Gli1* and *Gli2* gene expression were observed between the *nax* and wt cerebella (Figure 6C,D). *Gli3* mRNA expression also decreased in the wt, in a similar fashion as for *Gli1* and *-2*, but remained at the same level as on P5 at P10 and 15 in *nax* cerebella (i.e., *Gli3* expression was maintained, Figure 6E). As a result, expression levels were significantly different between wt and *nax* at P10 and 15. In addition, a heat map graphical analysis using RNAseq data sets from *nax* and wt cerebella at P5/P7 revealed that Shh signaling molecules are dysregulated in the *nax* cerebellum (Figure 7). For instance, *Shh* and its cofactors such as *Scube* are upregulated and *Smo* and *Grk2* appear downregulated in *nax* mice.

### 2.3. MYCN Expression Patterns Are Altered in the nax Cerebellum 

MYCN, a downstream target of the SHH pathway, is highly expressed in the embryonic brain and plays a vital role in development, especially regarding cerebellar GC precursor proliferation [29,30,31]. *Nax* and wt cerebellar sections were immunostained with an MYCN antibody at P5 and P12. Our results show that the protein localizes to GC precursors of the EGZ in both the wt (Figure 8A,C) and mutant (Figure 8B,D) and suggest a decreased immunoreactivity for MYCN in the *nax* cerebellum at P12 (Figure 8c,c’ vs. Figure 8d,d’). MYCN protein levels (Figure 8E) typically increase from P5 to P10, as shown for the wt, and subside towards the end of clonal expansion (P15). In the *nax* cerebellum, MYCN protein expression was comparable to wt at P5 but peaked earlier at P7 and then progressively decreased (Figure 8E). Importantly, MYCN protein expression was significantly lower in the *nax* mutant at P10 and 15. Both immunohistochemistry and Western blotting results suggest that MYCN protein expression is markedly decreased after cerebellar GC precursor proliferation peaked in the *nax* cerebellum and is virtually absent at P15. Next, we asked if protein downregulation in the *nax* cerebellum was due to decreased *Mycn* gene expression. Based on RT-qPCR results, no differences were observed in *Mycn* gene expression between *nax* and wt cerebella. RNA expression gradually decreased at P10 and 15 as compared to P5 to a similar extent in both groups (Figure 8F).

The observed reduced MYCN protein abundance could be due to an increased protein degradation or decreased synthesis. MYCN is degraded via the ubiquitin-proteasome system, which is regulated by catalyzing the polyubiquitination and degradation of MYCN to balance the proliferation and differentiation in GCs [47,48]. The proteasome expression was explored by immunostaining wt and *nax* MEF cells (MEF^wt^ and MEF*^nax^*) with anti-proteasome antibody. The proteasome immunoreactivity in both the nucleus and cytoplasm was apparently similar in MEF*^nax^* compared to MEF^wt^ cells. However, the proteasome activity assay demonstrated that the proteasome activity in MEF*^nax^* is significantly higher than in MEF^wt^, which may imply the potential for accelerated clearance of proteins such as MYCN (Appendix A). Further studies will be required to address this point.

It has been shown in MYCN-amplified neuroblastoma that the protein synthesis machinery is significantly upregulated [49]. A heat map graphical analysis using RNAseq data sets from *nax* and wt cerebella revealed that gene expression of both large (Figure 9A) and small (Figure 9B) subunits of the ribosomal proteins are strongly downregulated in the *nax* cerebellum. Of note, this observation might be associated with the strongly reduced number of GCs in the *nax* cerebellum.

## 3. Discussion

GCs constitute the largest neuronal population of the CNS. These neurons are significantly decreased in number in the *nax* cerebellar cortex (around 20% of that in wt controls). SHH signaling is important in the development of the GC precursors in the EGZ. SHH releases the inhibition of SMO by inducing a conformational change in the PTCH1. Derepressed SMO then activates the GLI factors and targets MYCN as well as cyclins, which drive GC clonal expansion in the EGZ [31]. 

In the present study, we describe the effects of the *Acp2* gene mutation on GC proliferation observed in the *nax* cerebellum. The total number of GCs is strongly reduced in the *nax* cerebellum, and PAX6 immunopositive GC precursors of the EGZ are dramatically decreased in number as well—particularly in the vermis—at P6, corresponding to the peak of GC precursor proliferation (Figure 2E,H,I). Because GCs rely on SHH signaling for their clonal expansion, we examined the expression levels of SHH as well as various other molecules involved in its signaling cascade as effectors or modulators during GC precursor proliferation in the EGZ. Our results indicate that SHH protein levels moderately increase over time (P5–19) and are comparable between PCs from *nax* and wt littermates. An increase in *Hhip* gene expression was observed in wt at P15, whereas levels remained low and comparable to P5 and 10 in the *nax* cerebellum. No relevant changes in comparison to wt controls were observed in the expression of mRNAs encoding the SHH receptor Ptch1 and the nuclear transducer Gli1. Interestingly, however, while no differential expression is detected in the levels of the *Mycn* transcript, the abundance of MYCN protein in whole cerebellar lysates is significantly lower in *nax* mice at P10 and, expectedly, remains much lower than in the wt animals at later stages (P15–19), given the low number of GCs in the *nax* cerebellum. 

Despite abundant neuronal degeneration in the *nax* cerebellum [39], it seems that the reduced number of GCs is due to the decreased proliferative activity of GC precursors during development (Figure 4). Numerous in vitro and in vivo studies have indicated that SHH is the primary mitogen that regulates the proliferation of GC precursors in the EGZ [16,18,21,50,51], and that SHH malfunction leads to reduced proliferative activity of the GC precursors [18,51,52]. After secretion by PCs, the effects of SHH on transcription factor GLIs in the target cells, in this case GCs, are co-regulated by *Hhip*, *Ptch1,* and *Smo*. A significantly lower cerebellar *Hhip* (HHIP is an SHH inhibitor) mRNA expression was observed in *nax* mice at P15, a stage where there is a decline in GC proliferation and clonal expansion is almost complete. How this paradoxical finding translates to HHIP protein expression and function needs to be further elucidated. It could be that because proliferation is already suppressed in the *nax* mouse, and fine-tuning inhibitory mechanisms that would result in an even larger reduction in proliferation, like those mediated by Hhip, are not being employed. Alternatively, HHIP expression has been suggested to be a readout of Shh signal transduction, like Ptc1 [53], and lower HHIP expression might therefore simply reflect decreased signaling, which is congruent with increased expression of the negative transducer in the pathway (i.e., Gli3) at these stages.

One of the readouts of SHH signaling pathway activity is transcription factor GLI1/2; from P5 to P15, *Gli1* and *-2* mRNA levels decreased and expression was comparable in the *nax* and wt cerebellar cortex at all time-points. Since the proportion of GCs is much smaller in the mutant, it is conceivable that individual GCs in the *nax* cerebellum express higher levels of Gli1/2, which would “compensate” for reduced cell number. Alternatively, Gli1/2 expression in Bergmann glial cells may also compensate for reduced expression in GCs, or it could be postulated that even though the number of total GCs is decreased in the *nax* cerebellum, this is proportionate to the size of the tissue, and that the consistent Gli1/2 expression indicates that GCs present in the *nax* cerebellum go through a relatively normal life. GLI1 belongs to the GLI family and participates in many cellular activities, such as cell proliferation and cell fate determination [24,25]. In the cerebellum, GLI1 is an important protein in GC precursor proliferation; as indicated, expression of *Gli1* and *2* was similar in the *nax* and wt cerebellum. Whereas *Gli3* mRNA levels decrease from P5 to P15 in the wt cerebellum, in a similar fashion as for *Gli1* and *-2*, *Gli3* expression (likely reflecting the PC layer) remains constant and is significantly higher at P10 and 15 in the *nax* cerebellum. It has been shown that in the absence of SHH signal, GLI3 converts to a repressor form and acts as a major negative transducer of the pathway [54]. Moreover, Gli3 expression is required for the Wnt-dependent negative regulation of SHH and control of cell proliferation and differentiation postnatally [55,56,57], and Wnt3 inhibits cerebellar GC proliferation [58], probably through Gli3. Importantly, ablation of GLI3 in *Shh* knockout mice restored the expression of several genes, including *Mycn* [59]. The sustained postnatal expression of *Gli3* after clonal expansion might represent a potential mechanism contributing to the reduced number of GCs in the *nax* cerebellum other than the SHH pathway.

*Mycn* gene expression was not significantly different in the *nax* cerebellum compared to wt littermate controls at any of the time-points studied. MYCN is an essential downstream effector of SHH signaling mediated by GLI during both normal and neoplastic cerebellar growth and is associated with GC precursor proliferation [31]. The *Mycn* gene is found amplified repeatedly in metastatic human neuroblastomas and serves as a transcription factor that participates in many cellular processes, including proliferation, differentiation, and apoptosis [31,60,61,62,63]. Impaired *Mycn* expression has been shown to disturb cerebellar development in mice as evidenced by a small size, disorganized and damaged foliation, and an 8-fold reduction of GC precursors compared with the control group [30]. In addition, in vitro and in vivo studies have demonstrated that *Mycn* is directly induced by SHH and that MYCN is required for GC precursor proliferation and plays a central role in SHH-mediated proliferation and tumorigenesis [31,64]. 

In contrast to *Mycn* gene expression, Western blot analysis showed that MYCN protein abundance is significantly decreased in the *nax* cerebellum; of note, this reflects protein expression after the clonal expansion peak. Proteasomal degradation of polyubiquitinated MYCN protein has been described [47,65]. Endogenous MYCN binds to ligases, such as HUWE1, prior to degradation by the proteasome [47]. Due to this degradation, MYCN is a short-lived protein with a half-life of only 30 min [5,31,61]. Degradation of MYCN withdraws the cells from proliferation and initiates their differentiation [47,65]. We observed preliminary evidence of increased proteasome activity in MEF*^nax^* cells. These findings suggest that in these fibroblasts grown from *nax* embryos, proteasome activity is increased, leading to augmented degradation of MYCN, which possibly explains reduced (GC) proliferation at a later stage of postnatal development. It is important to mention that in addition to proteasome impairment, it is conceivable that disruptions in processes such as cell recycling, autophagy, and responses to cellular stress contribute to our observations. Recently, we showed that the defect of lysosomal protein degradation in *nax* animals may be due to a malfunction in lysosomal activity. Thus, it is possible that our observed mismatch between gene and protein expression is associated with defect(s) of the proteasome system, autophagy, and/or lysosomal protein degradation in the *nax* model [66].

The observed alterations in the SHH-MYCN pathway in *nax* mice are preceded by a significant reduction in GCs during clonal expansion. At the peak of the clonal expansion, we observed Shh signaling pathway dysregulation. For instance, ligands (*Shh* and co-factors) are upregulated. and *Smo* and *Grk2*, key substrates and modulators of *Shh* signaling, are downregulated, in keeping with observations made by Zhao et al. [67]. Therefore, this evidence may suggest a likely upregulation of extracellular ligands due to disrupted Shh signals transduction, and a possible consequence is decreased MYCN expression. 

In addition, we have observed a downregulation of ribosomal protein gene expression in the *nax* cerebellum. MYCN is known to enhance the expression of a large set of genes functioning in ribosome biogenesis and protein synthesis [49]. While the GC proliferation defect may stem from downregulation of ribosomal protein expression, further investigation is required to assess the causal relationships between the two events in purified GCs. 

Cerebellar developmental tumors such as medulloblastoma are among the most common malignant pediatric brain tumors that originate from cerebellar GC precursor cells due to impairment of several molecular pathways, including SHH and MycN [48,68,69,70]. Our observations may pave the way for pharmacologic manipulation of *Acp2* as an approach of cerebellar developmental disorders such as the SHH group of medulloblastomas and Group 3 tumors and MYC signaling. 

## 4. Materials and Methods

### 4.1. Animal Model

All animal procedures described in this study were performed in accordance with institutional regulations and the *Guide to the Care and Use of Experimental Animals* from the Canadian Council for Animal Care and have been approved by local authorities “the Bannatyne Campus Animal Care Committee”, University of Manitoba (approved protocol # 15066). 

A *nax* mutant colony was established in the Genetic Modeling Center at the Faculty of Health Sciences, University of Manitoba, after the embryos were obtained from the Institute of Human Genetics at the University Medical Center, Georg-August University, Göttingen, Germany. The line was backcrossed with C57BL/6 breeders to maintain the *nax* mutation on a pure genetic background. The animals were kept at room temperature and relative humidity (18–20 °C, 50–60%) on a 12 hrs light–dark cycle. In this study, a total of 78 mice were used. After anesthesia (40% isoflurane, USP (Baxter Co., Mississauga, ON, Canada)), brains were removed and fixed in 4% paraformaldehyde (PFA) for immunohistochemistry (IHC) or prepared for Western blotting and real-time quantitative PCR. 

### 4.2. Genotyping

The *nax* mouse can be phenotypically distinguished from wild type (wt) littermates by appearance (small size, lack of body hair, and ataxia). In order to confirm the genotypes, we performed PCR using the following primers: Acp4F (5′ GCACTCTGTGCCTTCTCCAT-3′) and Acp4R (5′-CTGGGAGATTTGGGCAACTA-3′). After cleaving with the *Bam*HI restriction enzyme type II at 37 °C for 2 hrs, the DNA samples were loaded on a 2.5% agarose gel prepared in 0.5× TBE buffer with 1μg/mL ethidium bromide. A single 525 bp band identified the homozygous genotype; two bands (320 and 200 bp) identified the wild type genotype; three bands of 525, 320, and 200 bp identified the heterozygous genotype.

### 4.3. Cryosection

Whole brains were obtained from wt and *nax* mice and fixed overnight in 4% paraformaldehyde (PFA) at 4 °C. Next, PFA was replaced by sequential 10, 20, and 30% sucrose and kept overnight at 4 °C. The cerebellum was embedded in OCT compound and cryostat sectioned at 20 μm and processed for immunohistochemistry according to our laboratory SOPs (e.g., [71]).

### 4.4. Immunohistochemistry

For immunohistochemistry (IHC), we used our standard lab protocol, as previously published [38,39]. Briefly, the sections were washed with PBS and treated with 0.3% H_2_O_2_ to stop the endogenous peroxidase activity, followed by another wash. Sections were kept in blocking solution (10% normal goat serum and 0.05% Triton x-100 in PBS) for 1 h and subsequently incubated overnight at room temperature with primary antibody (anti-NeuN rabbit polyclonal, Millipore Sigma, Oakville, ON, Canada, 1:1000 [72]; anti-PAX6 mouse monoclonal, Developmental Studies Hybridoma Bank [73], 1:25; anti-calbindin D-28K mouse monoclonal, Swant, 1:1000 [39]; anti-sonic hedgehog rabbit polyclonal, Millipore Sigma, 1:200 [19]; anti-MYCN mouse monoclonal, Millipore Sigma, 1:25 [74], anti-proteasome 20S alpha + beta rabbit polyclonal, Abcam, Toronto, ON, Canada, 1:400, and anti-GFAP rabbit polyclonal (Millipore Sigma, 1:1000). After thorough washing with PBS, sections were treated for 2 h with the appropriate secondary antibody (goat anti mouse IgG, HRP conjugate, Millipore Sigma, 1:500; goat anti rabbit IgG, HRP conjugate, Millipore Sigma, 1:500). Diaminobenzidine (DAB) was used as a chromogen. For immunofluorescence staining, the secondary antibodies were Alexa Fluor 488 goat anti-mouse or anti-rabbit IgG and Alexa Fluor 594 goat anti-rabbit or anti-mouse IgG (Thermo Fisher, Ottawa, ON, Canada, 1:1000). For autofluorescence control, the primary antibody was omitted. 

### 4.5. Bromodeoxyuridine (BrdU) Incorporation Assay

BrdU (Sigma, Aldrich, Oakville, ON, Canada, Cat# 59-14-3) was dissolved in 0.007 N NaOH + 0.9% NaCl. At a dosage of 50 mg/kg body weight (~100 µL), BrdU solution was injected intraperitoneally to timed pregnant mice at E18 and pups were perfused at P5, or pups were injected postnatally at P6 and perfused after 45 min. The brain was removed and cryostat cerebellar sections were subjected to IHC using mouse monoclonal anti-BrdU (Sigma, Cat# B8434, 1:500).

### 4.6. Western Blotting 

Cerebellum samples were collected from *nax* mutants and wt littermates at different ages (P5, -7, -10, -15, and -19). The samples were placed in lysis buffer containing a protease inhibitor cocktail (Life Science, Peterborough, ON, Canada, Cat# M250) and phosphatase inhibitor (Sigma Aldrich, Oakville, ON, Canada, Cat# P5726). After sonication of the samples, protein concentrations were determined using a BCA protein assay kit (Bio-Rad, Mississauga, ON, Canada, Cat# 5000121). Protein samples were suspended in loading buffer (Tris-HCl 60 mM, glycerol 25%, SDS 2%, mercaptoethanol (ME) 14.4 mM, bromophenol blue 0.1%, H_2_O), and equal amounts of total protein (12 μL of each sample was loaded, 5 μL precision plus protein was used as marker standard (Thermo Fischer Scientific, ON, Canada)) were separated by sodium dodecyl sulfate polyacrylamide gel electrophoresis; based on the molecular weight of the protein of interest, 6–15% polyacrylamide gels were used. After separation, proteins were transferred to a polyvinylidene fluoride membrane. The membrane was blocked with 5% milk and incubated with primary antibody (Anti-Sonic hedgehog Rabbit Polyclonal, Millipore Sigma, 1:400; Anti-MYCN Mouse Monoclonal, Millipore Sigma, 1:200; in 1× TBST (Tris-buffered saline/0.01% tween 20) overnight at 4 °C, then washed 3 times with 1× TBST for 20 min each time and incubated with secondary antibody (goat anti mouse IgG, HRP conjugate, Millipore Sigma, 1:5000; goat anti rabbit IgG, HRP conjugate, Millipore Sigma, 1:5000) in 1× TBST for 2 h. Membranes were washed 3 times for 20 min with 1× TBST, and the immunoreactive bands were visualized by electrochemiluminescence (ECL) (Pierce, ON, Canada). The data were analyzed by AlphaEase software, and bands were normalized to β-actin expression.

### 4.7. RNA Isolation and Analyses

The cerebella of *nax* (P5; *n* = 2 and P7; *n* = 3) and wt littermates (P5; *n* = 3 and P7; *n* = 3) were isolated and total RNA was extracted with the RNeasy Plus Mini Kit (Cat# 74134, QIAGEN, Toronto, ON, Canada). RNA product concentration was measured by Nano-Drop ND-1000 UV-Vis Spectrophotometer (Thermo Fisher Scientific, Waltham, MA, USA). The samples were kept at −80 °C and sent to the McGill University and Genome Quebec Innovation Centre (MUGQIC). Raw RNAseq data were analyzed with FASTQ, and files were filtered and trimmed using the fastp package [75]. Next, cleaned data were aligned with the mouse reference genome (GRCm38.97) using Subjunc Aligner (v. 1.3.1) [76]; low-quality alignments, duplicates, and non-aligned sequences were removed from the aligned files using Samtools [77]. Finally, reads associated with each gene on the reference genome were extracted and counted using FeatureCounts software [78] and the “Reads Per Kilobase of transcript, per Million mapped reads” (RPKM) was calculated for each gene.

### 4.8. Gene Expression Analyses 

Total RNA was extracted using a TRIzol^®^ Plus RNA Purification kit (Life Technologies, Grand Island, NY, USA) according to manufacturer’s instructions. RNA quality and quantity were assessed by measuring the absorbance at 260 and 280 nm using a NanoDrop ND-1000 UV-Vis Spectrophotometer (Thermo Fisher Scientific, Waltham, MA, USA). All samples had an absorption ratio A260/A280 between 1.8 and 2.2. One microgram of RNA from each sample was treated with RQ1 RNase-Free DNase^®^ (Promega Corporation, Madison, WI, USA) to remove genomic DNA contamination, and reverse transcription was performed using SuperScript VILO cDNA Synthesis Master Mix (Invitrogen, Grand Island, NY, USA) according to manufacturer’s recommendations. Real-time quantitative PCR (RT-qPCR) reactions were performed in a Roche LightCycler 96 Real-Time System using Power SYBR green master mix (Life Technologies) in a final reaction volume of 20 μL; all samples were tested in triplicate. The stability of the housekeeping gene TATA Box binding protein (Tbp) was assessed as described previously [79,80]. Differences in the threshold cycle (ΔCt) number between the target genes and TATA Box binding protein (*Tbp*) were used to calculate differences in expression. 

### 4.9. Mouse Embryonic Fibroblast (MEF) Culture

Embryos were obtained from a heterozygous pregnant mouse. The tail was used for PCR to distinguish the genotype. The body was first minced and then digested with 0.25% Trypsin/EDTA (Gibco, Burlington, ON, Canada, Cat# 15090-046). After centrifuging the sample, the supernatant was discarded. The pellet was resuspended and added to a coated plate (0.1% Gelatin solution) and incubated with MEF medium (10% fetal bovine serum, 1% 200 mM l-glutamine, and 1% Penicillin-streptomycin in high glucose DMEM) at 37 °C and 5% CO_2_ until cells reached 100% confluency. The cells were immortalized with SV40 T antigen and then frozen and stored in liquid nitrogen. The MEF cells from wt (MEF^wt^) and *nax* (MEF*^nax^*) were collected, experimentally treated (if so required), and used for immunostaining, Western blotting, and proteasome activity assays.

### 4.10. Proteasome Activity Assay 

To analyze 26S proteasome activity in MEF cells, we measured the proteolytic activity of 20S and the deubiquitinating activity of 19S using fluorescent substrates. For more detailed information, refer to recent publications by Abu-El-Rub et al., 2019 and 2020 [81,82]. Briefly, 100 μL of the Proteasome Lysis Buffer (Cayman chemical, Ann Arbor, Michigan, USA, Item No. 10011097) was added to MEF cells, followed by incubation with gentle shaking for 30 min at room temperature. After centrifugation, 90 μL of supernatant from each sample was transferred to a black 96-well plate, followed by the addition of 10 μL of SUC-LLVY-AMC proteasome substrate (Cayman chemical Item No. 10011095) and 10 μL 26S assay buffer. The fluorescent intensity was assessed to determine 20S subunit proteolytic activity, after 1 h of incubation in the dark (excitation = 360 nm; emission = 480 nm). The deubiquitinating activity of 19S was measured by transferring 90 μL of each sample to a 96-well black plate, and 0.5 μL of the Ubiquitin-Rhodamine 110 (Boston Biochem Item, Toronto, ON, Canada, Cat# U-555) proteasome substrate solution and 10 μL of 26S assay buffer was added to each sample; the fluorescent intensity for 19S subunit activity was measured immediately after 1 h (excitation = 485 nm; emission = 535 nm). 

### 4.11. Imaging

A Zeiss Axio Imager M2 microscope (Zeiss, Toronto, ON, Canada) was used to capture bright field images, which were analyzed with Zeiss Microscope Software (Zen Image Analysis software; Zeiss, Toronto, ON, Canada). Fluorescent images were captured by Zeiss Z1 and Z2 Imagers and a Zeiss LSM 700 confocal microscope (Zeiss, Toronto, ON, Canada). Adobe Photoshop CS5 was used to edit, crop, and finalize the images for presentation. 

### 4.12. Cell Counting

Cerebellar GCs in the granular layer and GC precursors in the external germinal zone were counted using ImageJ software. Three slides were counted for each wt and *nax* cerebellum. Three sections were counted on each slide based on the standard size field (950 µm × 715 µm). 

### 4.13. Statistical Analysis

GraphPad Prism 6 was used to evaluate the statistical significance of differences by analysis of variance (ANOVA) for more than two groups and student *t*-test for comparison of two groups. Differences were considered to be statistically significant when *p* < 0.05 (level of significance is indicated in the figures (* *p* < 0. 05, ** *p* < 0.01)).

## 5. Conclusions

SHH-MYCN signaling has been shown to promote the proliferation of GC precursors in the cerebellum. In the cerebella of *nax* mice, the number of GCs and GC precursors amount to only ~20% of that in the wt littermates. In this study, we showed that cerebellar MYCN protein expression is markedly reduced in *nax* mice at around P10, while *Mycn* gene expression remains unchanged. Upstream, the observed sustained postnatal expression of *Gli3* in *nax* cerebella might affect SHH signaling, as it is also almost aligned with the temporal pattern of changes in MYCN expression. At the peak of GC proliferative activity, Shh signaling pathway is dysregulated and the ribosomal protein gene expression is reduced in the *nax* cerebellum, suggesting an impairment in other molecules associated with GC proliferation. Additional mechanisms regulating GC development during the clonal expansion period are likely impaired and warrant further investigation.

## Figures and Tables

**Figure 1 ijms-22-02994-f001:**
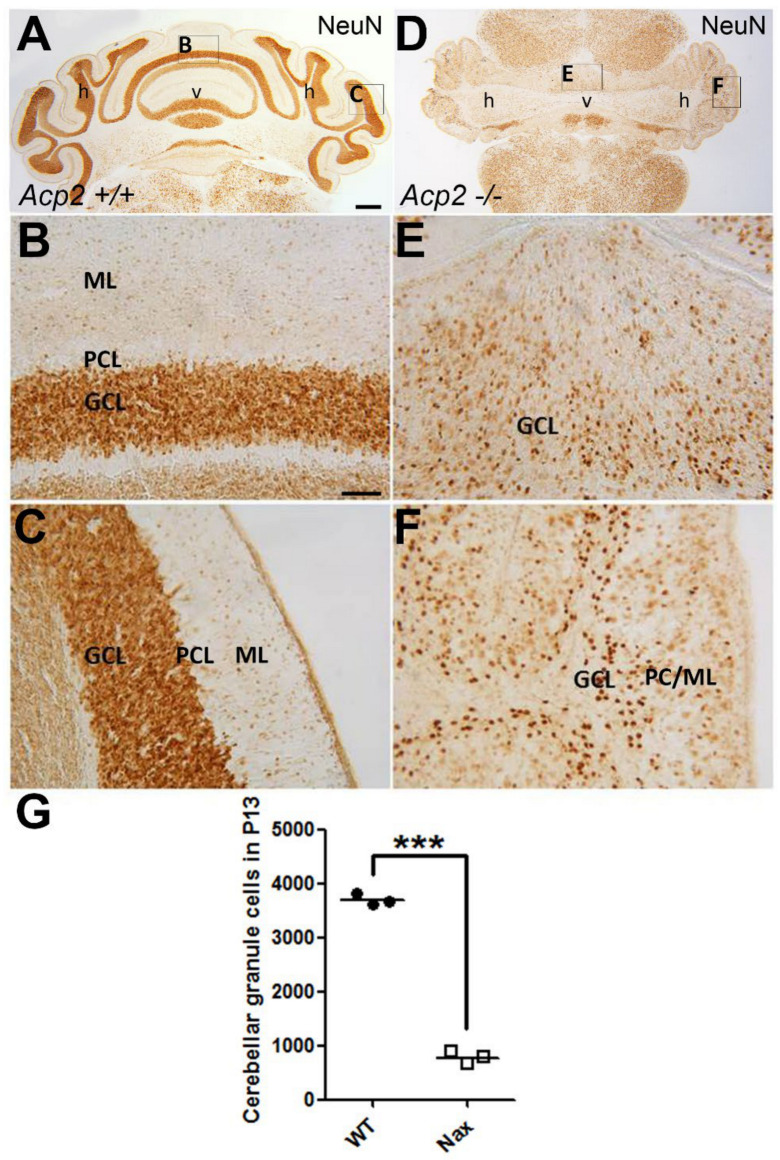
The number of cerebellar granule cells (GCs) is significantly decreased in *nax* compared to wt littermate mice. (**A**–**C**) Frontal section of P13 wt mouse cerebellum immunostained with neuro nuclei (NeuN) reveals a large number of mature GCs in the GC layer (GCL). The higher magnification shows medial (**B**) and lateral (**C**) views of the wt cerebellum (**A**). (**D**–**F**) Frontal section of P13 *nax* mouse cerebellum immunostained with NeuN shows that the cerebellum size is smaller and the lobules are underdeveloped. Evidently, the number of cerebellar GCs is much lower (**E**,**F**) than in the wt cerebellum ((**B**,**C**) for direct comparison). The higher magnification shows medial (**E**) and lateral (**F**) views of *nax* cerebellum (**D**). (**G**) Cell counting of the frontal sections of P13 samples shows a significant difference in the number of GCs between wt and *nax* cerebella (**G**). The graph represents the average cell counts at P13 in sections from wt and *nax* littermates (wt; *n* = 3 and *nax*; *n* = 3) and shows that the number of GCs amounts to only ~20% of that scored in wt cerebella. The data in the graph are presented as the mean ± SEM. Statistical analysis was performed using unpaired *t*-test (*** *p* < 0.001). Abbreviations: GCL, GC layer; PCL, PC layer; ML, Molecular layer; h, hemisphere; v, vermis. Scale bars: 500 μm in (**A**) applies to (**A**,**D**), and 100 μm in (**B**) applies to (**B**,**C**,**E**,**F**).

**Figure 2 ijms-22-02994-f002:**
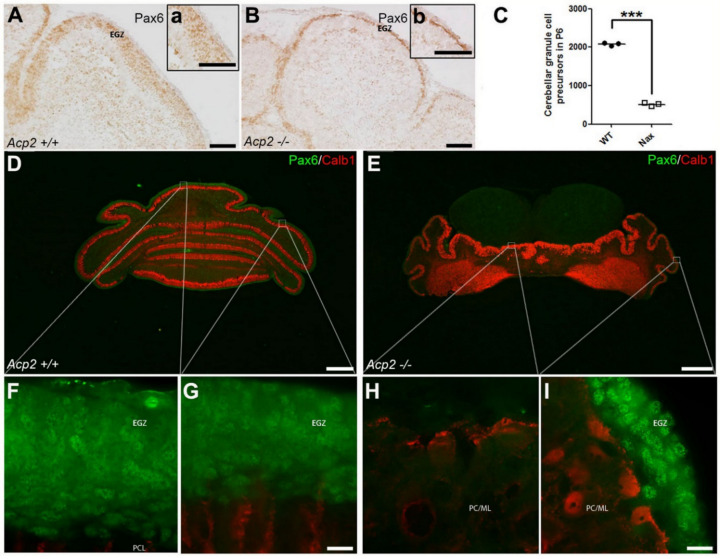
The number of cerebellar GC precursors is decreased in the *nax* mouse. (**A**) Sagittal section of P6 wt mouse cerebellum immunostained with PAX6 shows GC precursors in the external germinal zone (EGZ); higher magnification provided in (**a**). (**B**) Sagittal section of P6 *nax* mouse cerebellum immunostained with PAX6 reveals that the number of GC precursors is markedly lower than in the wt cerebellum; higher magnification provided in (**b**). (**C**) Cell counting of sagittal sections of P6 cerebellum shows a significant difference in the number of GC precursors during early postnatal age between wt and *nax*. The bars represent the average cell counts of cerebellar sections obtained at P6 from wt and *nax* littermates (wt; *n* = 3 and *nax*; *n* = 3) and show that the number of GC precursors in *nax* is only about 20% of that scored in wt. The data in the graph are presented as the mean ± SEM, and statistical analysis was performed using unpaired *t*-test (*** *p* < 0.001. (**D**–**I**) Frontal sections of wt (**D**) and *nax* (**E**) cerebellum at P8 immunostained with PAX6 (green) and CALB (red). (**D**) In the wt cerebellum, a thick external germinal zone (EGZ) could clearly be observed medially (**F**) and laterally (**G**). (**E**) In the *nax* mutant, the number of proliferating GCs was strongly reduced in the EGZ as compared to the wt; GCs appeared completely absent medially (**H**)**,** and only a thin EGZ could be identified laterally (**I**). Abbreviations: EGZ, external germinal zone; ML, molecular layer; PCL, Purkinje cell layer. Scale bars: 500 μm in (**A**,**B**), 100 μm in (**a**,**b**), 1000 μm in (**D**,**E**), 20 μm in (**G**) applies to (**F**–**G**), and 20 μm in (**I**) applies to (**H**,**I**).

**Figure 3 ijms-22-02994-f003:**
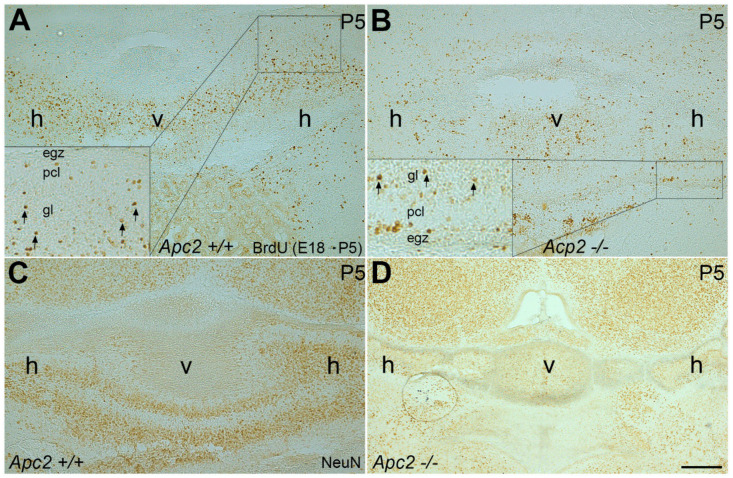
Visualization of BrdU incorporation in frontal sections at P5, following BrdU injection at E18 in timed pregnant mice. (**A**,**B**) BrdU staining shows that immunopositive cells migrate from EGZ to the GC layer (gl) in the wt (**A**) and *nax* (**B**) cerebellum, which is indicated by arrows in the inset. (**C**,**D**) NeuN immunostaining of serial sections of wt (**C**) and *nax* (**D**) cerebella confirming migration of mature GCs to the gl. Abbreviations: egz, external germinal zone; gl, granule cell layer; h, hemisphere; pcl, Purkinje cell layer; v, vemis. Scale bars: 200 μm in (**D**) applies to (**A**–**D**).

**Figure 4 ijms-22-02994-f004:**
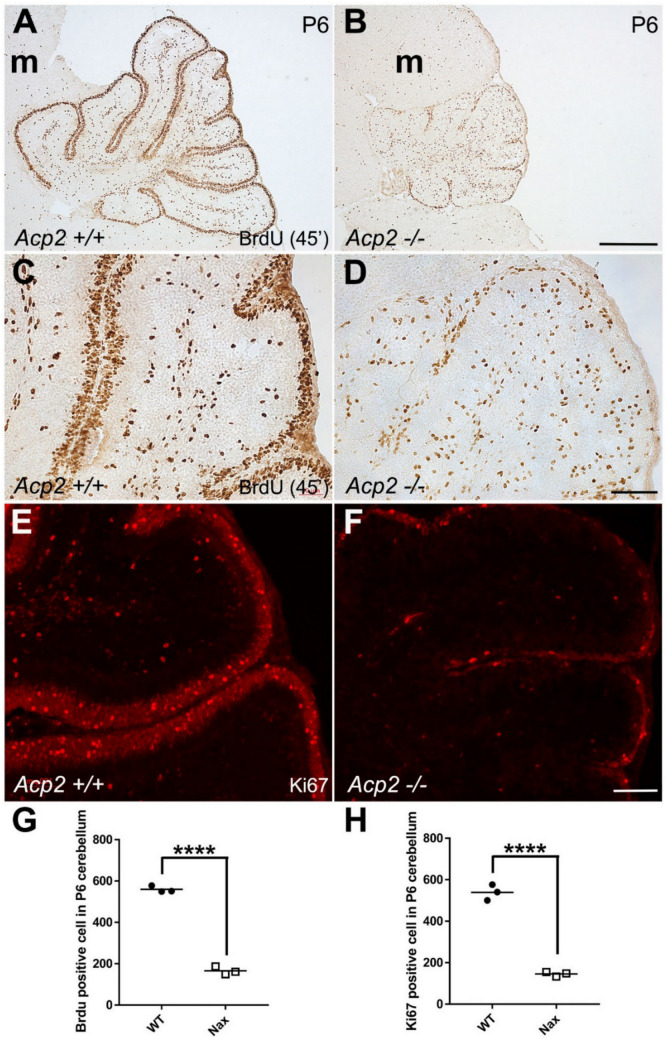
Using BrdU and ki67 to assess GC proliferative activity in wt and *nax* cerebella at P6. (**A**–**D**,**G**) Visualization of BrdU incorporation in *nax* and wt cerebellar sagittal sections, 45 min after BrdU injection at P6. BrdU staining revealed a higher number of proliferating (immunopositive) cells in the EGZ in wt ((**A**), higher magnification in (**C**)) as compared to *nax* cerebellum ((**B**) and higher magnification in (**D**)). Number of proliferating cells in the EGZ that are BrdU immunopositive is significantly higher in wt than in *nax* littermates (wt; *n* = 3 and *nax*; *n* = 3). Quantitative analysis shows that the BrdU^+^ index is about 30% in *nax* compared to wt. (**E**,**F**,**H**) Ki67 immunostaining of cerebellar sections confirmed that the number of proliferating cells in the EGZ is higher in wt (**E**) than in *nax* mice (**D**). Similar to the BrdU results, cell counting of sagittal sections at P6 shows a significant reduction in the number of Ki67 positive cells in the EGZ in the *nax* cerebellum than in wt littermates (wt; *n* = 3 and *nax*; *n* = 3). The data in the graph are presented as the mean ± SEM, and statistical analysis was performed using unpaired *t*-test (**** *p* < 0.0001). Abbreviation: m, mesencephalon. Scale bars: 500 μm in (**B**) applies to (**A**,**B**), 100 μm in (**D**) applies to (**C**,**D**), 100 μm in (**F**) applies to (**E**,**F**).

**Figure 5 ijms-22-02994-f005:**
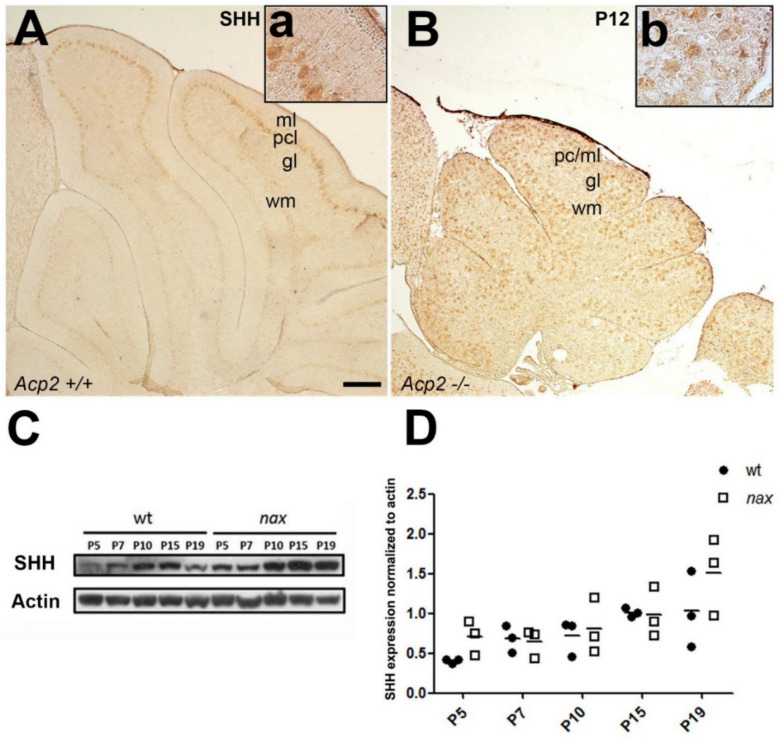
SHH is expressed to a similar extent by cerebellar PCs in wt and *nax* mutant mice. (**A**,**B**) Sagittal sections of P12 wt and *nax* mouse cerebella immunostained with SHH show that the expression of SHH is localized in PCs in both groups. (**C**,**D**) Western blot analysis of SHH expression at P5, -7, -10, -15 and -19 reveals a modest increase over time and similar expression levels in wt and *nax* cerebella at all time-points (wt: *n* = 3 and *nax*: *n* = 3). The actin shows consistent and equal loading of all cerebellar samples. The data in the graph are presented as the mean ± SEM, and statistical analysis was performed using one-way ANOVA. Abbreviations: gl, GC layer; ML, molecular layer; PCL, PC layer; wm, white matter. Scale bars: 100 μm in (**A**) applies to (**A**,**B**).

**Figure 6 ijms-22-02994-f006:**
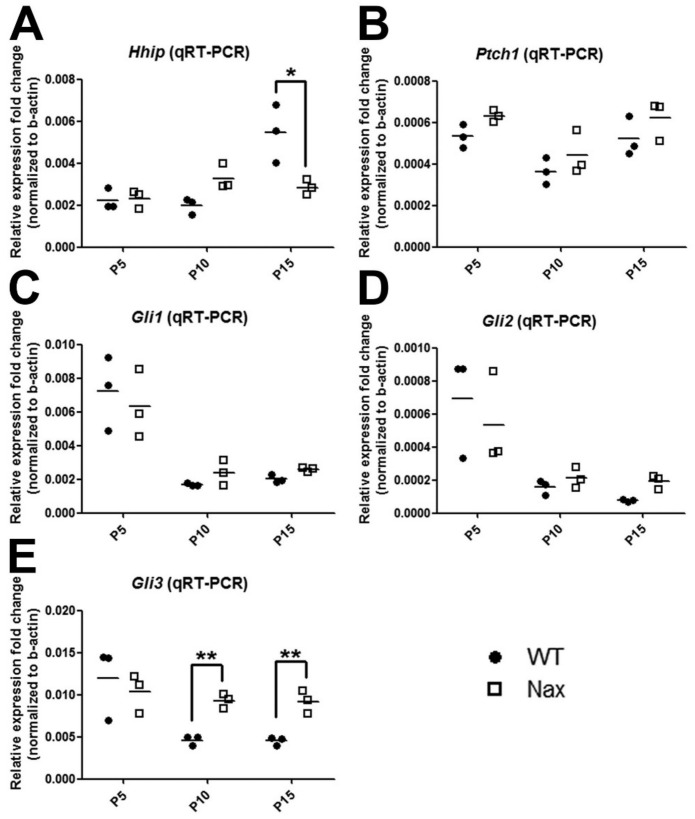
mRNA transcription levels of *Hhip*, *Ptch1**,* and *Gli1–3* in wt and *nax cerebella*. (**A**) Based on RT-qPCR, *H**hip* gene expression in wt and *nax* cerebella was comparable at P5 and P10, which are the most critical days for GC proliferation. In wt mice, expression appears to be increased at P15, whereas it remained at the same level in *nax* animals (* *p* < 0.05 compared to wt). (**B**) No differences in *Ptch1* expression were observed between wt and *nax* cerebella; expression levels did not significantly change over time (P5–15). (**C**) As compared to P5, *Gli1* gene expression decreased over time at P10 and 15; no significant differences were observed between the *nax* and wt cerebellum at any time point. (**D**) *Gli2* mRNA expression decreased over time in a similar fashion as for *Gli1*; no significant differences in *Gli2* expression were observed between the *nax* and wt cerebellum. (**E**) *Gli3* mRNA expression decreased over time in the wt cerebellum, in a similar fashion as for *Gli1* and *-2*, but was maintained high (same level as on P5) at P10 and 15 in the *nax* cerebella. Expression levels were significantly different between wt and *nax* at P10 (** *p* < 0.01) and 15 (** *p* < 0.01). The data in the graph are presented as the mean ± SEM, and statistical analysis was performed using an unpaired *t*-test.

**Figure 7 ijms-22-02994-f007:**
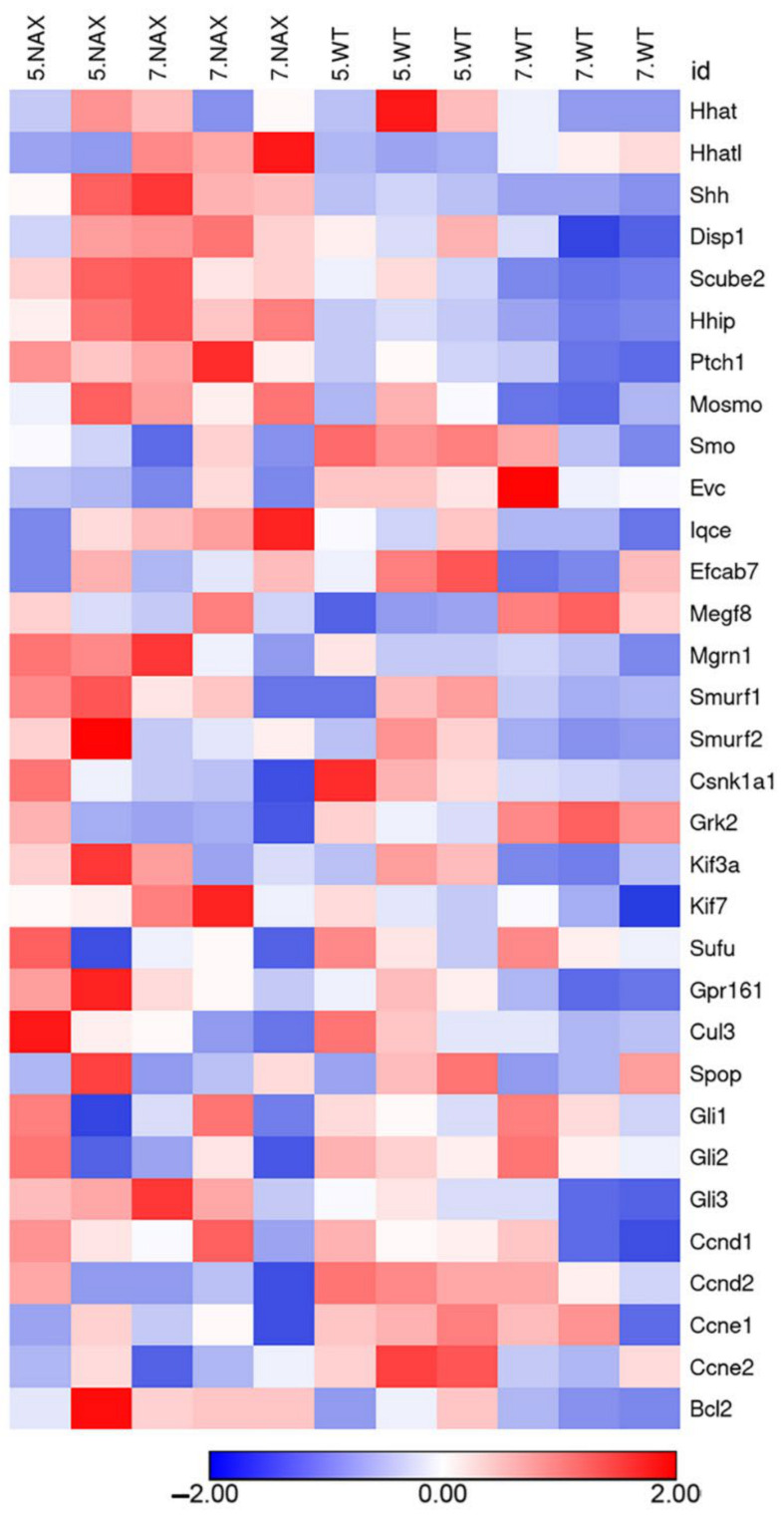
Heat map analysis of SHH pathway in the *nax* cerebellum at P5/P7. The rows of the heat map represent the genes of interest and the columns represent the experimental samples. Each cell is colored based on the level of expression of that gene in that sample. This heat map analysis shows color-coded expression levels of up (red)- and down (blue)-regulated genes. The data show that SHH signaling pathways are dysregulated in the *nax* cerebellum.

**Figure 8 ijms-22-02994-f008:**
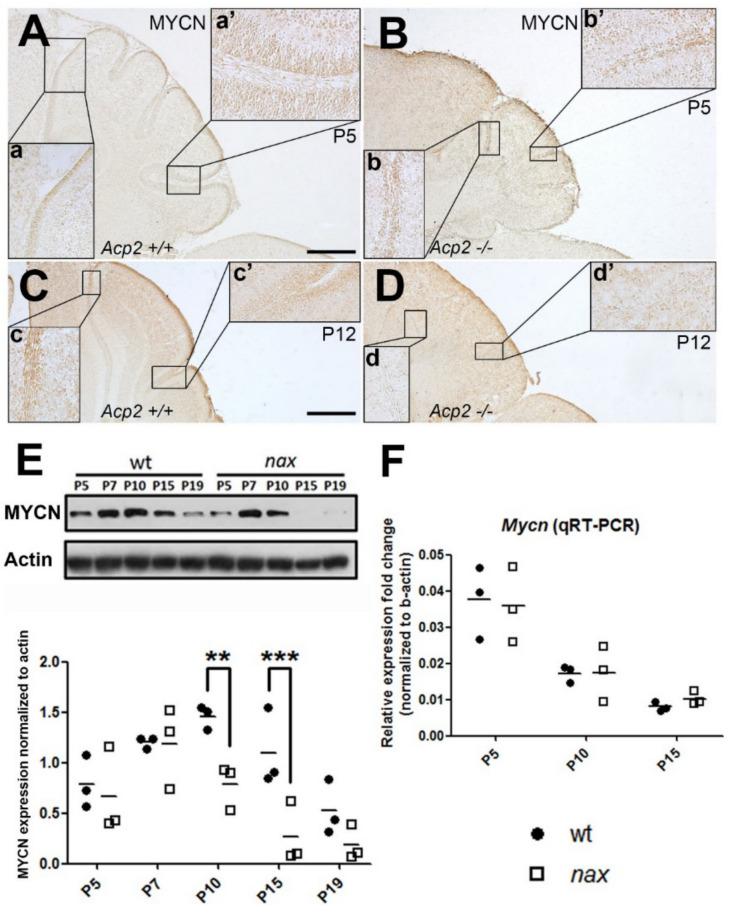
MYCN is expressed by GC precursors in wt and *nax* cerebella, but cerebellar MYCN protein levels are lower in the *nax* mutant mouse at P10 and P15. (**A**,**B**) Sagittal sections of P5 wt and *nax* mutant mouse cerebella immunostained with MYCN show expression in GC precursors. Higher magnifications of the wt (**A**) and *nax* (**B**) anterior (**a**,**b**) and posterior (**a’**,**b’**) lobes are shown in the inserted panels. (**C**,**D**) Sagittal section of P12 *nax* mutant mouse cerebellum (**D**) immunostained with MYCN shows lower expression in GC precursors compared with wt littermates (**C**). Higher magnifications of the wt (**C**) and *nax* (**D**) anterior (**c**,**d**) and posterior (**c’**,**d’**) lobes are shown in the inserts. (**E**) Western blot analysis of MYCN expression in wt and *nax* cerebellar lysates at P5, -7, -10, -15 and -19 (wt: *n* = 3 and *nax*: *n* = 3). In the wt samples, MYCN protein levels increased from P5 to P10 and subsided towards the end of clonal expansion (P15). In the *nax* cerebellum, MYCN protein expression was comparable to wt at P5, but peaked earlier at P7 and then progressively decreased. Importantly, expression was significantly lower in the *nax* mutant at P10 and P15. The actin shows consistent and equal loading of all cerebellar samples. The scatter plots show individual data points and statistical analysis was performed using one-way ANOVA (** *p* < 0.01 and *** *p* < 0.001). (**F**) Cerebellar *Mycn* mRNA expression gradually decreased at P10 and 15 as compared to P5 to a similar extent in both *nax* and wt mice. No significant differences in *Mycn* expression were observed between the groups at any time point. β-actin gene expression was used to normalize the qRT-PCR data. The data in the graph are presented as the mean ± SEM, and statistical analysis was performed using an unpaired *t*-test. Scale bars: 500 μm in (**A** applies to (**A**,**B**) and 500 μm in (**C** applies to (**C**,**D**).

**Figure 9 ijms-22-02994-f009:**
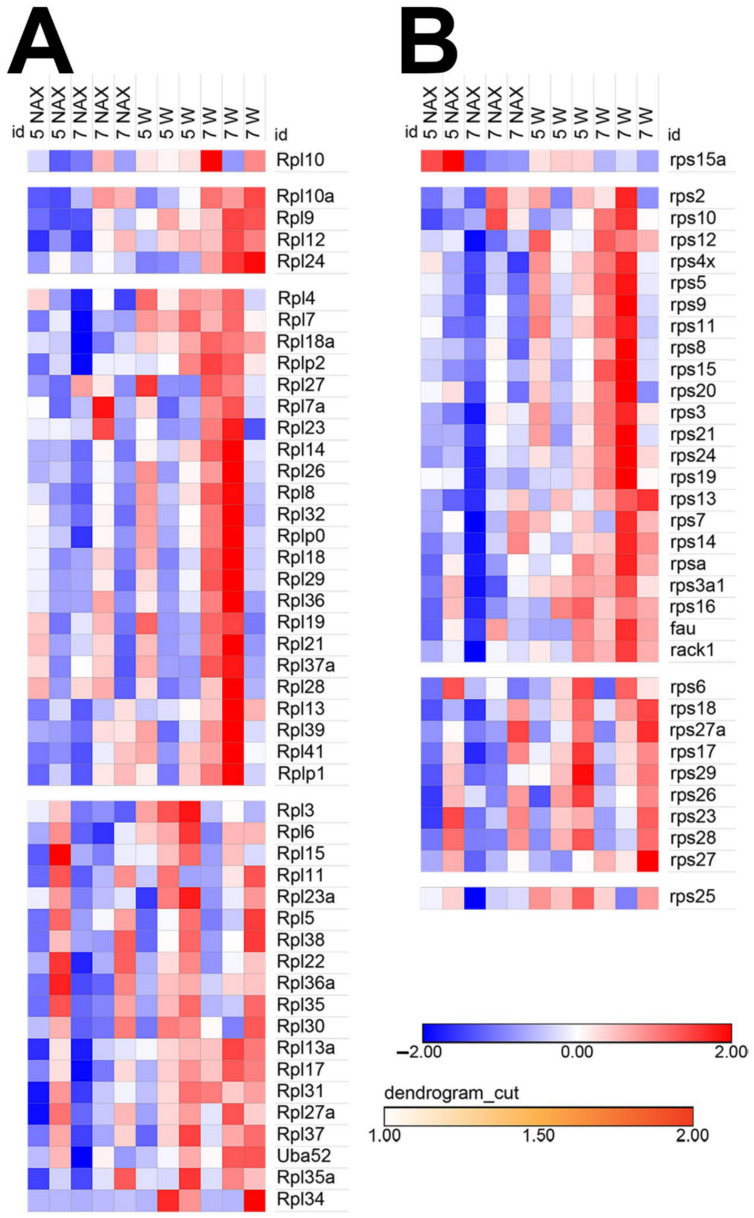
Heat map analysis of ribosomal protein gene expression in the *nax* cerebellum at P5/P7. The rows of the heat map represent the genes of interest and the columns represent the experimental samples. Each cell is colored based on the level of expression of that gene in that sample. This heat map analysis shows color-coded expression levels of up (red)- and down (blue)-regulated genes. The data show that major clusters of genes coding for large (RPL), (**A**) and small ribosomal subunits (RPS), (**B**) are markedly downregulated in the *nax* cerebellum.

## Data Availability

Data is contained within the article or Appendix A.

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
