# Peer review of "Reduced Granule Cell Proliferation and Molecular Dysregulation in the Cerebellum of Lysosomal Acid Phosphatase 2 (ACP2) Mutant Mice"

_ijms, 2021, doi:10.3390/ijms22062994_

Round 1
Reviewer 1 Report
The manuscript of Jiao and colleagues describes the analysis of cerebellar defects in a mutant mouse model of Lysosomal acid phosphatase 2 (Acp2). The paper is quite intriguing as recently other Authors (Canterini et al., 2017; Costa et al., 2017) have described Shh signaling defects in other experimental models of lysosomal disorders. In general the manuscript, particularly in the firt part, appears sound and clear. However, there are major flaws and weakening issues related to the mechanistic analysis, that have been depicted in the second part.
Major issues:
1) Starting from Figure 2A-B, images are often of poor resolution.
This applies particularly to Figure2A-B, 3 (whole) and Figure 7(A-D). Authors should provide better quality images and magnifications.
2) Authors showed decreased proliferation (BrDU and Ki67) levels in the cerebella and reduced Pax6 immunostaining. Double immunostaining for PAX/BrdU or Pax6/Ki67 would be encouraged to support the hypothesis of decreased granule cell precursors proliferation.
3) Authors hypothesize that a Shh signaling defect could upstream induce the c-MYC decrease. It's likely possible, but as neither the ligand (Shh), nor Gli1 (a downstream target) are decreased this appear at odds with their hypothesis. My suggestion would be a temptative rescue with an agonist in vivo, or more simply by an in vitro assay on isolated cerebellar cells (see below)
4) Authors have performed their analysis of the proteasome system in MEF. But all data were previously collected on isolated cerebella (RQ-PCR and western). Why Authors have not performed their proteasome assays in in vitro cultures of isolated cerebella ? The use of the in vitro cultures would enable to answer also to the previous question (3) related to Shh signaling:by providing a Shh agonist Authors could show that indeed the impaired Shh signaling is responsible to the cell proliferation defects. It might be possible by this way also that Authors could measure c-MYC proteolysis by an anti ubiquitin antibody on c-MyC immmunoprecipitates. I would strongly encourage these analysis.
5) Lastly, Authors just showed briefly the heat map of RNA seq data on their mice discussing the decrease of Ribosomal proteins. Were also targets of the Shh pathway downregulated in their RNA seq. data?. How would they relate the decrease of protein synthesis (more general defect) to a limited Shh signaling impairment. it might be possible that also Wnt signaling could be affected, as well other pathways, including autophagy. In summary the RNA seq data seem to point to a general defect, which would mean that the Shh pathway itself may be just one of the widespread defective pathways. Authors should revise their last observation accordingly (Fig.8) and provide more details about their RNA seq data. (also in Supplementary files). If other pathways appear dysregulated the whole paper (including the title) should be then rephrased.
Reviewer 2 Report
The paper by Jiao et al describes rare mechanisms in acid phosphatase 2 pathway.
The manuscript is well written; all experiments design and results presented are clear. I am not able to assess whether similar projects have been published on a similar topic.
My only suggestion is that the authors could elaborate in the usefulness of their findings in the context of clinical presentation/manifestation, new therapies and clinical finings. Have these results re-volutionalized medicine?
It would be worth elaborating on it in the discussion section.
Round 2
Reviewer 1 Report
After carefully reading the rebuttal letter, I personally understand the covid-19 related situation and, therefore, I assume that the experiments which I requested cannot be done. However, the Supplementary Fig.3 provided by the Authors, appear to strenghten the hypothesis of a Shh dysregulation. Either smoothened and grk2 appear downregulated in the RNA-seq data. Both are key substrates and modulators of Shh signaling. It appears indeed that the ligand (Shh) and its cofactor (Scube) are upregulated in nax mutant mice. Therefore, there is a likely upregulation of extracellular ligands due to disrupted Shh signaling transduction.
In my opinion, Authors should emphasize the data of Supplementary Fig.3, either by including them as a main Figure or stressing along the text (particularly in the Discussion) that either smo or grk2 are dysregulated (in agreement with Zhao et al., 2016 ). Probably the analysis of these targets (by RQ-PCR or Western) would have made the conclusions more robust.
On the other hand I would suggest to remove Fig.8 or tune down the role of protein degradation as this mechanism may have little or partial relation to the more specific dysregulation of the Shh pathway and the likely consequent decrease in cMYC immunoreactivity.
